# The Use of Artificial Sputum Media to Enhance Investigation and Subsequent Treatment of Cystic Fibrosis Bacterial Infections

**DOI:** 10.3390/microorganisms10071269

**Published:** 2022-06-22

**Authors:** Aditi Aiyer, Jim Manos

**Affiliations:** Charles Perkins Centre, Infection, Immunity and Inflammation, School of Medical Sciences, The University of Sydney, Sydney 2006, Australia

**Keywords:** artificial sputum media, cystic fibrosis, narrative review, opportunistic infections, biofilm

## Abstract

In cystic fibrosis (CF), mutations in the CF transmembrane conductance regulator protein reduce ionic exchange in the lung, resulting in thicker mucus, which impairs mucociliary function, airway inflammation and infection. The mucosal and nutritional environment of the CF lung is inadequately mimicked by commercially available growth media, as it lacks key components involved in microbial pathogenesis. Defining the nutritional composition of CF sputum has been a long-term goal of in vitro research into CF infections to better elucidate bacterial growth and infection pathways. This narrative review highlights the development of artificial sputum medium, from a viable in vitro method for understanding bacterial mechanisms utilised in CF lung, to uses in the development of antimicrobial treatment regimens and examination of interactions at the epithelial cell surface and interior by the addition of host cell layers. The authors collated publications based on a PubMed search using the key words: “artificial sputum media” and “cystic fibrosis”. The earliest iteration of artificial sputum media were developed in 1997. Formulations since then have been based either on published data or chemically derived from extracted sputum. Formulations contain combinations of mucin, extracellular DNA, iron, amino acids, and lipids. A valuable advantage of artificial sputum media is the ability to standardise media composition according to experimental requirements.

## 1. Introduction

### 1.1. The Cystic Fibrosis Sputum Microenvironment

Cystic fibrosis (CF) is an autosomal recessive disorder caused by congenital mutations in the cystic fibrosis transmembrane conductance regulator (CFTR) gene [1,2]. The CFTR is a member of the ATP-binding, cAMP-dependent, phosphorylation-activated anion channel superfamily of proteins that transport chloride and bicarbonate across the apical plasma membrane (APM) of epithelial cells [3]. The CFTR maintains the hydration of secretions within airways by managing ionic transport across epithelial cells. Additionally, the CFTR modulates the activity of the epithelial sodium channel (ENaC) among other ion channels [4]. CFTR dysfunction in the lung results in thicker mucus which impairs ciliary function, alters mucus pH and mucus stasis, and impairs innate immunity due to airway inflammation [3]. (Figure 1: CFTR dysfunction in cystic fibrosis bronchiole).

The mucus itself is a key indicator of lung health. It is a gel layer, which in healthy states comprises approximately 97% water and 3% solids (mucins, salts, lipids, and cellular debris). Mucins are large, highly anionic glycoproteins linked by hydroxyl side groups to sugar chains containing tandemly repeating sequences of serine and threonine rich in amino acids [5]. While many different mucin products have been identified, two polymers, MUC5AC and MUC5B, are highly expressed in the mucus of human airways (Figure 2: Mucus hypersecretion).

MUC5AC is produced by surface goblet cells in the proximal airways, whereas MUC5B is produced by both secretory cells and submucosal glands [8]. These homotypic polymeric mucins are continuously synthesised to replenish the gel layer. They create a mucin mesh by cross-linking and entanglement via non-covalent calcium bonding [7]. The glycan side chains are then able to bind large quantities of liquids which give the mucus its lubricant character and the ability to be effectively cleared by mucociliary action and coughing [9]. Conversely, CF mucus becomes dehydrated due to lack of ionic exchange, and this is compounded by the upregulation of MUC5AC and MUC5B gene expression, which increases solid content from 3% in healthy mucus to 15% in CF mucus, resulting in dehydration of the mucus layer and insufficient clearance by mucociliary action [10,11]. These changes foster a suitable ecological niche for opportunistic infections by a variety of pathogens [12,13]. Initial respiratory tract infection with pathogens in a planktonic state is often followed by formation of microbial biofilms, eventually culminating in recurrent and subsequently chronic infections [12,13,14].

### 1.2. Modelling the CF Lung Mucus Environment

The mucosal and nutritional environment of CF lung has been poorly defined and inadequately mimicked by commercially available laboratory growth media [15]. While it has been previously asserted that the host itself provides nutrients for sufficient bacterial growth, evidence is mounting that mucus in the CF lung is key in microbial pathogenesis [16]. Instead, it is host-provided nutrients that mediate colonisation and disease. Thus, it is critical to define the nutritional composition of CF sputum to better elucidate bacterial growth and infection pathways in vitro.

CF creates conditions that promote polymicrobial colonisation in varied oxygen environments. Bacteria may grow as an aggregate and, depending on the strain, can eventually form a biofilm within the high mucus environment of the CF lung. Additionally, facultative aerobes can also grow deep within the hypoxic regions of deep mucus, providing evidence that the environment is microaerophilic [17]. While the CF lung is predisposed to colonisation by a variety of opportunistic bacterial pathogens, the most common is *Pseudomonas aeruginosa*. After lung infection, this Gram-negative bacterium initially grows as free-swimming cells in the CF lung airway surface liquid (ASL). The impaired clearance of foreign cells from the CF airways is conducive to the eventual formation of clusters of microcolonies and progressively to the formation of biofilms in the hypoxic zones of the airway lumen, a feature of chronic infection [17].

### 1.3. Artificial Sputum Media: A Lab-Replicable Medium with CF Sputum Roots

Understanding the CF lung microenvironment compared to that of a healthy lung, and the basic nutritional requirements of bacteria, form the foundation for creating a biologically relevant medium. One factor that influences lung homeostasis and infection is pH. The average pH of submucosal gland fluid in CF patients is approximately 6.6–7.0. However, it is possible that the pH of these secretions could be different in vivo. Physiological comparison using in situ pH measurements of airway mucus obtained from explanted airways was carried out by Palmer et al. (2007), who compared it to submucosal gland fluid and found that both samples had similar pH values (between 6.0 to 6.9) [15].

### 1.4. The Significance of Respective Components in Artificial Sputum Media

Within a normal lung, the major macromolecular components of mucus are extracellular DNA (eDNA), filamentous actin, lipids, proteoglycans and MUC glycoproteins. By contrast, CF lung sputum contains little intact mucin and has an increased concentration of all the above macromolecular components. Mucin has been shown in studies to be a key niche for bacterial microcolony and biofilm formation in vivo, while also providing the inherently high viscosity environment characteristic in CF [18].

The presence of aromatic amino acids has been shown to induce both antimicrobial activity and differential expression of iron-regulating genes during bacterial growth in CF sputum [19]. During a study by Kumar and Cardona, the motility of *B. cenocepacia* was observed to increase in response to the addition of five amino acids: arginine, glutamate, histidine, phenylalanine, and proline, when grown in synthetic CF medium (SCFM) [20]. Additionally, bacteria tend to move towards carbon energy sources, as evidenced by chemotaxis and amino acid utilisation [21]. Therefore, the increased amino acid concentration has a two-fold effect on *B. cenocepacia*: it contributes to increased motility via flagellar synthesis and results in a general chemotaxis towards preferential energy sources.

From the evidence presented, it can be concluded that the inclusion of amino acids in artificial sputum media may enhance bacterial biofilm growth by contributing to bacterial metabolite production and subsequent strengthening of the exopolysaccharide biofilm matrix [22]. While few of these experiments have been conducted in artificial sputum, the addition and removal of certain amino acids may also be applied to biofilms grown in ASMDM as a way of establishing the best growth conditions. Studies in the past decade have indicated that amino acid concentrations should be sustained in an almost homeostatic balance to appropriately contribute to biofilm growth. This “homeostasis” is maintained by the two chiral varieties of amino acids, L and D amino acids (L-AA and D-AA, respectively). L-AA is generally used in ribosomal synthesis while D-AAs serve a dual role, by enhancing cell wall and protein synthesis but by also destabilising cell wall structures, thus impeding bacterial growth [23]. These studies are quite subjective, being reliant on the characteristics of the bacterial species and individual genetic virulence mechanisms. However, increasing the concentration of D-AA has been shown to disrupt bacterial biofilm synthesis both alone [24] and in combination with antibiotics [25]. Thus, amino acids are important in maintaining bacterial secondary metabolite production and subsequent biofilm formation [20]. Additionally, during any infection exacerbation episodes, the overall amino acid concentration has been demonstrated to increase in correlation to the severity of the pulmonary disease [5].

A major contributor to bacterial biofilm growth in artificial sputum media is viscosity as this mimics the viscous properties of CF mucus. In vivo, the presence of extracellular DNA (eDNA) within *P. aeruginosa* biofilms can provide necessary viscosity. In a study by Haley et al. (2012), the importance of eDNA was demonstrated via induction of biofilm formation in a lowered oxygen environment. *P. aeruginosa* and *S. aureus* were incubated under static conditions in artificial sputum medium with lowered oxygen tension (10% Environmental O_2_), mimicking the conditions observed in the CF lung [18]. In vivo, the sources of eDNA are dead host immune cells, lysed bacteria, or Quorum sensing (QS)-regulated release of bacterial DNA. In their study, they increased the eDNA concentration threefold from 0.5× (2 mg/mL) to 1.5× (6 mg/mL) eDNA to assess the implications of higher concentrations of eDNA. While lowered concentrations of eDNA resulted in well-developed PAO1 biofilms, the space they occupied was considerably small [19]. The increased concentrations of eDNA did not translate to bigger and more visible biofilms; instead, the eDNA formed a gelatinous mass with scattered individual cells [18]. Thus, any variations in the delicate balance of eDNA levels affected the formation of biofilm, such as structures more dramatic than the levels of mucin.

The surfaces of the CF lung are consistently exposed to a high oxygen concentration. This increases the potential for oxidation of unbound, atmospheric iron, eventually resulting in catalytic formation of reactive oxygen species (ROS). Airway cells are required to achieve rapid iron homeostasis via uptake of non-protein bound iron by divalent ion transporters on the apical surface of bronchial epithelial cells [26]. However, in the CF lung, there is a natural genetic impairment in the function of cells with a dysfunctional CFTR, and sputum from CF patients contain micromolar concentrations of iron, making this micronutrient more readily available to inhaled pathogens [19,27]. Considering that iron is an essential bacterial nutrient sensed by quorum signalling, for biofilm structural integrity, maintaining the correct balance of iron is critical to sustaining optimal bacterial physiology. When supraphysiological iron concentrations are increased from the normal 20–25 μM (<1 μM of which is protein bound), the environment is more conducive to growth of biofilms that contain less eDNA, leading to lowered levels of microcolony formation and increased bacterial susceptibility to antimicrobials [28]. Within artificial sputum, most formulations include an iron chelator. While this may at first seem counterintuitive, the impaired iron homeostasis observed in CF epithelial cells means that the introduction of a chelator is to regulate the amount of free iron in the media, especially in the event of excess iron being introduced inadvertently in commercially available additives to ASM. As outlined in Neve et al. (2021), most ASM formulations use commercial porcine gastric mucin (PGM) as their mucin source [29]. From their analysis of the components of commercial PGM, it was identified as a source of contaminating bioavailable metals, lipids, and amino acids. Thus, this imbalanced ionic and amino acid content needs to be regulated within the delicately balanced ASMDM using the chelator, diethylenetriaminepentaacetic acid (DTPA).

Human sputum contains a high concentration of lipids. In an analysis of cystic fibrosis sputum samples by Sahu and Lin in 1978, lipids constituted approximately 30% of dry material [30]. The main types were neutral lipids, such as triglycerides and cholesterol. In a study by Harmon et al. (2010), mucus production and retention may be regulated by lipid binding molecules. Thus, the interconnectivity between mucus production and lipid content, specifically via 15-keto-prostaglandin-2 (15-keto-PGE_2_) to its receptor PPAR-γ, is influential in cystic fibrosis pathogenesis [31].

In theory, the most physiologically relevant method to study CF infections would be to use sputum obtained directly from CF patients. However, patient sputum is not consistent in composition, even within a gender, age group or exacerbation level. It is also inefficient and impractical due to changes in consistency in sterilisation practices. When considering patient-to-patient differences, there is an increase in mucin glycoproteins during infection exacerbations. Thus, patient-derived sputum would be highly variable and affected by patient’s infection stage at the time of collection [32]. To compensate for this high variability and inefficiency, artificial sputum medium has been created to mimic the average composition of CF sputum and to provide a replicable, biologically relevant formulation that may be used for testing of genetic expression and potential therapeutic interventions accurately in vitro.

### 1.5. The Evolution of Artificial Sputum Media

Artificial sputum media have been developed in an attempt to replicate the environment of in vivo infection in the cystic fibrosis lung, to monitor microbial physiology, gene expression and the efficacy of different treatments. The earliest iteration, developed in 1997, focused on the two main components that form mucus, mucin and DNA sources. Neve et al. (2021) outlined nine distinct formulations that have been created, and these may be differentiated by their varied phenotypes and metabolic profiles, as discussed below. The formulation may be roughly separated into two lineages, literature-based and CF sputum-derived. (Figure 3: Timeline of artificial sputum medium formulation).

The first category is based on studies indicating or establishing the concentrations of nutrients commonly found in CF sputum. The foundational artificial sputum formulation in this category is that of Ghani and Soothill (1997), which is based on concentrations of nutrients most found in CF sputum reflected in the literature and contains mucin from porcine stomach [34]. This base formulation was further modified in subsequent studies by the addition of specific amino acids, bovine serum albumin (BSA)-altered concentrations of mucin and extracellular DNA, and the replacement of amino acids or iron source. Key nutrients present in this category are lecithin, provided by egg yolk emulsion, and an iron chelator, DTPA.

An entirely synthetic sputum medium with precise concentrations of ingredients based on the microanalysis of extracted CF sputum has also been developed [15]. The benefit of using analysis-based specifications is the ability to mimic the CF sputum environment; however, the comparison was made with actual sputum, where sample-to-sample variation and collection methodology likely diminished the overall accuracy of comparison. The converse is true for literature-based formulations where mirroring “real” samples is lacking, but the consistency of evidence collected via research is a positive factor.

The evolution of sputum formulations, especially in the literature-based category, does not necessarily follow a linear trend. Instead, the nature of the formulation is adapted according to experimental requirements. This highlights the inherent mutable nature of artificial sputum and its capacity to be adjusted.

The sputum media should contain all ingredients needed to satisfy the nutritional requirements of bacteria, not only to grow, but to also form visible biofilm structures. The Ghani and Soothill formulation, however, lacked the second capacity. The cultures remained turbid, as in any growth medium, but lacked the conventional polymeric matrix seen in when growing *P. aeruginosa* biofilms [33]. This may be attributed to the absence of amino acids which contribute to secondary metabolite production and biofilm formation.

Therefore, an improvement on the original formulation was observed in the Sriramulu et al. (2005), with the addition of an individual amino acid sources and an iron chelator. This formulation delivered the most overt phenotypic presentation when the PAO1 strain was grown, and thus, it has been used in various other studies, as it provides the most basic and consistent model in which to conduct biofilm-based experiments [22]. Sriramulu et al. (2010) followed up on their formulation by making modifications based on the recorded average amino acid concentration in CF lung (5 mg/mL) and accomplished this by adding a combination of individual amino acids to a final concentration of 250 mg/mL. When the medium was inoculated with *P. aeruginosa*, microcolonies were able to form within the ASM+ media and not on the abiotic well surface, as evidenced by crystal violet staining [5]. In *P. aeruginosa* biofilms specifically, the expression of alginate has been shown to contribute to the exopolysaccharide and thus mature biofilm formation. The expression of alginate correlates with tighter microcolony formation. This finding suggested that free amino acids could contribute to adaptive bacterial evolution and chronic infection.

The ASM published by Kirchner et al. in 2012 is compositionally identical to the Sriramulu (2005) formulation except for its enhanced buffering capacity and the way that it has been sterilised via filtration. This is to prevent any denaturation or destruction of mucin and amino acid components [17]. The use of the ASMDM media allowed for micro-aerophilic to anaerobic conditions conducive to bacterial growth which would eventually form biofilms within the sputum components [17].

The study by Fung et al. (2010) produced an artificial CF sputum medium (ASMDM) based on modifications of ASM+ [22] that more closely approximate the sputum found in CF patients in terms of concentrations of the components and physical properties identified in the literature. The Sriramulu (2010) base-formulation was altered by adding BSA, increasing mucin concentration, and reducing eDNA levels, while the source of amino acids was changed from individual stocks to Casamino acids [32]. The Casamino acid combination notably lacks asparagine, cysteine (Cys), glutamine (Gln), and tryptophan, which are notably higher in CF sputum. However, it is more important to have an adequate overall concentration of amino acids versus specific amino acids. Another modification made in ASMDM was the addition of 10 mg /mL BSA which reflects higher albumin concentration in CF sputum due to vascular leakage associated with CF lung inflammation [38,39]. For these reasons, ASMDM may better mimic the hypoxic or anaerobic environment of the CF lower-airway mucus plugs and is therefore more characteristic of a post-CF exacerbation episode. A unique addition to this formulation is 1% agarose to increase its viscosity with the goal of better visualisation of microcolony appearance and size. The Fung et al. study found that *P. aeruginosa*, an opportunist commonly found in CF airways, grew normally and deeply invaded the ASMDM, suggesting that ASMDM mimicked the CF lower airway mucus where deep growth of *P. aeruginosa* under hypoxic conditions has been identified [32]. The strains used included UCBPP-PA14, AES-1R and two clinical CF isolates. When the transcriptional profile of *P. aeruginosa* grown in ASMDM was compared to that of the growth in CF sputum, Fung et al. demonstrated that bacterial nutrient availability and uptake, carbon consumption and cell–cell signalling profiles were similar for *P. aeruginosa* in both media.

The formulation of Hare et al. (2012) is derived from the ASMDM of Fung et al. However, was modified by adjusting the buffering capacity of ASMDM, which was reduced, and the DTPA was replaced with ferritin to better replicate CF sputum iron availability [35]. These changes fostered a better environment to study proteome changes in *P. aeruginosa* AES-1R, the acute infection CF strain when compared with its chronic counterpart AES-1M. Their results were unique compared to previous studies, as they were focused on the phenomenon of higher iron availability in the CF lung [40,41] and the subsequent siderophore upregulation observed in *P. aeruginosa* strains [42]. Thus, their formulation could be used to assess a similar phenomenon in other opportunists and in mixed cultures and was later used in a study of glutathione disruption of *P. aeruginosa* biofilms by Klare et al. (2016) [43].

The Quinn formulation is the latest iteration of the original ASM. Developed in 2015, the formulation dubbed “ASMRQ” was used to study physiological fitness of CF microbes based on the principles of the Winogradsky column (WinCF system). The WinCF system was developed to study the chemical, sediment, and oxygen-based stratification of microbial communities, similar to what would be observed in the CF lung where microbial communities form stratified biofilms. Their sputum reverted to the Fung formulation, with the added modifications of removing BSA and 1% agarose, and increasing the porcine mucin concentration to 20 mg/mL, to better reflect their experience with CF sputum consistency [36]. Another key change from the Hare and Fung formulations was the substitution of minimal essential medium amino acids (MEM AA) in the place of Casamino acids. While MEM AA lacks Cys and Gln, this change was made with the intention of maintaining amino acid availabilities in the body [44].

Nutritional changes in ASM create distinct changes in the phenotypic presentation of the bacteria, evidenced by changes in iron and amino acid content and concentration. As highlighted earlier, the addition of amino acids alone was able to create visible difference in ASM in comparison to Ghani and Soothill, in that a biofilm was able to be visualised. When observing the Sriramulu, Kirchner, and Fung formulations, it should be noted that all of them lacked iron sources which possibly impacted *P. aeruginosa* phenotypic presentation. The lack of an iron source did not prevent growth or biofilm formation but created a dense tan-green biofilm at the air-liquid interface, with large open pores within the structure giving way to turbid growth toward the bottom of the well. With the addition of iron in the Hare formulation, the phenotype changed from the green-coloured biofilm to a dense, brown-coloured biofilm without the same structural features as its predecessors. The phenotypic changes must, therefore, be induced by changes in secondary metabolites.

An alternative to literature-based formulations is the development of a defined formulation derived from chromatic and enzymatic analyses of CF sputum samples collected from patients. This concept was applied in 2007 by Palmer et al. in the creation of the first synthetic sputum medium (SCFM1), testing its efficacy using *P. aeruginosa* strain UCBPP-PA14 and *Staphylococcus aureus* strain Xen 36 [15]. Considering the complexity of sputum, replicating its contents as accurately as possible is difficult, especially when considering the natural variation which exists between samples [15]. While a standardised synthetic formulation is near impossible to achieve, SCFM still provides a substrate with carbon energy sources to support high density microbial growth, seen in chronic CF colonisation [45].

SCFM1 was developed from extracts of sputum samples from 12 adult CF patients who had not suffered an exacerbation. These samples were assessed following a chromatographic and enzymatic analysis, specifically their anion concentrations using anion-exchange chromatography and free amino acid levels. These data were exclusively used to develop SCFM, independently of previously developed literature-based formulations. In essence, the media were developed from average concentrations of ions, free amino acids, glucose, and lactate to create a high amino acid content media. While the formulation by Palmer et al. (2007) lacks mucin, which is undefined, its advantage is the ease with which its components and their concentrations can be adjusted. When the gene expression profiles of SCFM1 and extracted CF sputum were compared, to assess nutrient availability and whether bacteria could source similar levels of nutrients to sustain growth in both media, their carbon consumption and cell–cell signalling profiles were highly similar when testing *P. aeruginosa*. SCFM1 was used in antimicrobial assays by growing bacteria at 37 °C for 18 h prior to coinfection with exponential growth phase *S. aureus*. Combined with the increased quorum-sensing molecules and increased aromatic amino acids in SCFM, *P. aeruginosa* was able to competitively lyse CF lung cohabitant *S. aureus*, which highlights the importance of amino acid content in contribution of virulence. While this has been observed in *P. aeruginosa*, it is yet to observed with other CF species [15].

One caveat regarding these studies with SCFM1 lies in the inherent lack of mucin, which provides a high viscosity environment for biofilm microcolony structures to form. However, this is rectified by adjusting mucin concentrations according to experimental requirements when performing biofilm assays. Another matter to consider is the nature of samples collected; that is, the patients all had non-exacerbating CF. CF presents in various stages, with no or low-level exacerbations being more common in younger patients [46]. Exacerbations are associated with increased sputum production, decreased airway function and the acquisition or change in microflora [46]. This suggests that SCFM1 is less representative of many CF types. However, this is not an inherent flaw, but instead highlights the fact that SCFM1 may be formulated to mimic different stages of CF mucus based on sample collection. This could provide insights into the nutritional cures that drives bacteria colonisation at each stage and the requirements for an optimised treatment strategy.

To improve upon the shortcomings, SCFM2 and SCFM3 were developed by the addition of components that were lacking in SCFM1. SCFM2 contained free DNA from salmon sperm, bovine maxillary mucin, *N*-acetylglucosamine (GlcNAc), and dioleoyl phosphatidylcholine (DOPC) at concentrations similar to those present in CF sputum [37]. To assess how closely SCFM2 approximates the nutritional capacity of authentic CF sputum, *P. aeruginosa* strains’, PAO1 and PA14, genetic fitness levels were assessed in SCFM2 via transposon insertion sequencing. It became evident that a specific set of genes responsible for anabolic pathways for biosynthesis of thiamine, nicotinamide adenine dinucleotide (NAD), purines, folate, and branched amino acids and tryptophan metabolites was higher in CF sputum than in SCFM2. Therefore, the final iteration of the SCFM series (SCFM3) was modified by the addition of *p*-aminobenzoic acid, NAD+, adenine, guanine, xanthine, and hypoxanthine to compensate for their absence in the previous formulation and to address the abundant bioavailability of these metabolites in CF sputum [33].

## 2. Uses of Artificial Sputum Media in the Study of CF Infections

While artificial sputum media were developed initially to provide a replicable medium in which to model CF infections, over time, its applications have become more extensive. Beyond the initial goal of identifying bacterial gene expression, artificial sputum uses now include screening for phage therapy and understanding differences in gene expression in acute and chronic infection isolates, to providing a medium for an in vitro lung cell model.

By uncovering shifts in genetic expression in terms of virulence factors between acute and chronic CF isolates, this information could be used to screen the efficacy of artificial sputum in replicating the CF microenvironment. Naughton et al. used acute and chronic isogenic strains of *P. aeruginosa* (AES-1R and AES-1M, respectively) to identify significant changes in gene expression and loss of expression in these isogens collected over 10 years apart from the same patient [47]. By using ASMDM. they were better able to model the differences between acute and chronic strains in a replicable CF in vitro environment. Previous work has focused on gene expression in the *P. aeruginosa* reference strains PAO1 and PA14, but in this study, Australian clinical epidemic strains were used, which allowed the detection of expression of AES-1 genes not found in the reference strain. This work highlighted the fact that the use of the ASMDM sputum media provides a more holistic profile of differential expression genes under sputum-like conditions [47]. Their objective was to utilise the AES-1R and AES-1M isogen sequences taken from the same patient at acute and chronic infection and a “PANarray” containing eight non-redundant sets of *P. aeruginosa* gene probes to identify gene expression differences between early and chronic stages of disease pathogenesis by AES-1. The study identified the chronic isogen (AES-1M) as having undergone significant upregulation of alginate expressing genes in ASMDM, suggesting that this was triggered by the CF sputum-like medium, as chronic infections have been characterised by increased alginate expression, which enhances biofilm production and resistance to leukocyte killing [48]. By comparison to the acute AES-1R, AES-1M demonstrated upregulation of certain virulence-related genes for biofilm enhancement and increased resistance, and this upregulation has not been evident when acute and chronic isolates grown on other media have been compared. This study can be extended in ASMDM to assess QS molecules in their ability to downregulate virulence gene expression in the ASMDM.

ASMDM has also been used to conduct gene expression studies in tandem with treatments against biofilms. In a study by Klare et al. (2016), 72 h biofilms of clinical CF isolates, including AES-1 strains and Liverpool Epidemic strains (LES), in ASMDM+ (+indicates addition of ferritin) were treated with the antioxidant glutathione (GSH) and DNase I. Bacterial counts of ASMDM+ growth were enumerated after eight hours of treatment, allowing time for biofilm disruption to occur. The Naughton et al. study monitored gene expression in all strains and found that there was a significant upregulation of pyoverdine biosynthesis genes in GSH-treated biofilms in ASMDM [43]. Increased pyoverdine synthesis combined with the low iron levels in the ASMDM media indicate greater biofilm structural formation. Considering the importance of iron in pyoverdine synthesis, this provides evidence that iron chelation could enhance the efficacy of GSH treatment by preventing biofilm reformation [49].

Another branch of study using artificial sputum in CF infections goes beyond differences in genetic expression profiles and focuses on the treatment of the bacterial cells. The first stage in treatment of opportunistic bacterial infections in CF is determining the presence of specific bacterial species in the lungs and their antimicrobial susceptibility. This is generally performed using a minimum inhibitory concentration (MIC) and compared against pre-determined cut-offs. The caveat in this is that predetermined MIC cut-offs are based on aerobic planktonic bacteria grown in a medium that does not mimic the CF lung mucus. To overcome the issue of planktonic cultures in aerobic conditions being used to determine MIC, Kirchner et al. (2012) created a microtitre plate assay to determine both planktonic and ASMDM-biofilm-grown cell minimum metabolic inhibitory concentrations using resazurin as a metabolic indicator. Using 15 different clinical isolates of *P. aeruginosa* and a PAO1 reference strain and a variety of LES strains with six extra non-LES strains, their study focused on testing susceptibility to tobramycin, an aminoglycoside antibiotic commonly used to treat CF and other lower respiratory tract infections. The results of this study were apparent in the ASMDM-grown *P. aeruginosa* and suggested that tobramycin may be less effective, which reflects previous studies using conventional testing methodologies [17]. A key difference between commercially available traditional growth media is that cultures grown in ASMDM thrive despite reduced oxygen concentrations, and this has been shown to alter the behaviour of *P. aeruginosa* and change its antibiotic susceptibility. The use of ASMDM, considering its cost effectiveness and comparability to conventional testing methodologies, should not only be applied to other clinical isolates of *P. aeruginosa* but to other opportunistic infections in CF.

Another aspect to consider in the treatment of bacterial infections is the implications of increased antibiotic resistance. Antibiotic treatment alone, particularly if repeated, is prone to creating resistant species, especially if treatments are impeded by the viscous sputum environment. Thus, more novel antimicrobials and antibiofilm agents are being tested, and their efficacy, as with antibiotics, needs to be assessed in a biologically relevant medium.

A combination of the Kirchner and Sriramulu artificial sputum media was used by Kosztolowicz et al. (2020) to provide a sputum-based environment to study antibiotic transport through a *P. aeruginosa* biofilm. Biofilms are significantly more complex, and recent studies have shown that assessment of potential therapeutic interventions using predetermined planktonic MIC cut-offs does not mimic biofilm MICs.

Kosztolowicz et al. hypothesised that antibiotic particles may interact with biofilm microcolonies scattered throughout the artificial mucus background in a “plum pudding” model, and this would match with experimental outcomes [50]. This sputum, as previously outlined, is a highly complex medium with a gel-like consistency. Therefore, determining physicochemical interactions between bacteria and diffused antibiotics is difficult if using a conventional growth medium that lacks the composition of ASM. The Kosztolowicz study used a diffusion model to both theoretically and practically show the movement of the antibiotic ciprofloxacin through ASM using pre-determined temporal functions. Their experimental diffusion setup model was conducted by forming PAO1-mature biofilms for 96 h at 37 °C in ASM and quantifying the percentage of the membrane covered by PAO1 biofilm using crystal violet staining. Following this, the antibiotic ciprofloxacin was diffused through the medium to determine a concentration gradient in one dimension. A hypothesised theory behind the interaction between antibiotics and sputum components could be binding to mucin in sputum, thus reducing free drug levels [51], and this would affect diffusion in addition to the development of a biofilm. Their results indicated that bacteria could be forming a diffusion barrier as part of their natural defence. This was shown, as the ciprofloxacin measured diffusion constant was delayed in biofilm dense regions. While this method of measuring the diffusion bactericidal effect in a physiologically relevant medium may be useful in cystic fibrosis treatment, there are more complex bacterial defences that when investigated would provide more insight into the efficacy of treatment.

In a study by Hall et al. (2019), the antibacterial action of nitric oxide (NO) was tested using a custom-made diffusion cell. NO is an endogenously produced free radical capable of inducing oxidative stress to damage and kill bacteria without resulting in associated bacterial resistance. By comparing the diffusion coefficients of NO through 1× Phosphate Buffered Saline (PBS) to a CF-relevant, proteinaceous ASM developed from the literature-based solutions by Kirchner and Sriramulu, they were able to quantify how the complexity of mucus impacted the movement of free NO when compared to standard mucus-free media [52]. As expected, diffusion through the proteinaceous ASM was slower than through PBS and could be attributed to two factors: physical obstruction by mucin glycoproteins and/or protein scavenging. Therefore, while NO is a promising bacteriocidal agent in vitro [53], by using CF-relevant media, it has been highlighted that biological complexity impedes its potential for antibacterial action, as it would be difficult to achieve minimum inhibitory concentrations. Conversely, the diffusion cell methodology used in this manner would be ideal to test nebulised therapies in the future, as it may allude to the nature of antibacterial action of the compound tested when assessed in a biologically relevant medium.

In the age of emerging antibiotic resistance, an alternative to antibiotics is bacteriophage therapy, using bioengineered bacterial viruses (bacteriophages) to provide a targeted therapy without the high likelihood of resistance development. In the study by Garbe et al. (2010), sputum media were used to test this newer avenue of treatments [54]. The inherent advantages of bacteriophages are their specificity against bacteria; however, their application requires not only an understanding of the bacteria but a holistic consideration of the in vivo conditions. It has been documented that phages which multiply well under in vitro conditions could well fail to do so in vivo; thus, an appropriate in vitro model must be employed to mitigate this potential issue. The study by Garbe et al. tested the ability of the phage JG024 to lyse a non-mucoid wild-type strain of *P. aeruginosa*, PAO1, by monitoring both phage particles and alginate production, the latter of which is a well-documented response to oxygen-limited conditions and a potential indicator of infection. By using the Sriramulu (2010) [5] artificial sputum medium instead of the conventional LB medium employed by most studies, the authors were able to deduce that *P. aeruginosa* responded as expected in a chronic infection setting by overproduction of alginate and that this alginate is what hindered the efficacy of activity of the phage [54]. The development and testing undertaken in artificial sputum provided a feasible model in which they were able to investigate the possibility of phage-lysing bacteria under comparable infection conditions as would be observed in vivo.

Another interesting use of artificial sputum media has been in the testing of nanocarrier-mediated transmucosal drug delivery. This proposed treatment is dependent on muco-adhesive, -inert or -penetrating nanoparticles diffusing efficiently through the mucus in the cystic fibrosis lung to optimise drug delivery in the lung where there is a high mucus content. Artificial sputum medium was utilised by Nafee et al. (2018) to study the colloidal stability and turbidimetric function of the nanoparticles as they moved through the medium, especially as their efficacy is contraindicated in high-viscosity conditions [55]. Thus, the author’s primary objective was to identify if the nanoparticles could pass effectively through the mucus barrier. Nanoparticle penetration through the ASM was observed via 3D time-lapse, and a clear diffusion gradient over time was observed. These findings could lead to future research focused on nanoparticle delivery of treatments.

A limitation of artificial sputum is that while it is a good surrogate for CF sputum, it is not an exact replica. It lacks the high viscosity and rheological behaviour observed in CF sputum. When CF sputum was substituted, it seemed to slow down the diffusion of nanoparticles. While it may seem preferable to conduct experiments in extracted CF sputum, the issues described above with respect to lack of heterogeneity amongst sputum samples means that obtaining reproducible results is challenging. Therefore, ASM was considered a suitably consistent substitute for modelling the diffusion of nanoparticles.

An interesting novel application of the medium has been the creation of a feasible model for bacterial infection that includes host cells submerged in artificial sputum. Wijers et al. (2016) investigated the effectiveness of ASMDM superimposed on a monolayer of human A549 lung carcinoma cells, as a dual-component model to investigate *Burkholderia cenocepacia* infections in CF [56]. Wijers et al. modelled their study on a previous one by Sajjan et al. (2004) who had investigated the presence of *B. cenocepacia* on well-differentiated human CF cells and in non-CF human lung epithelial cells without the addition of ASMDM [57]. They identified a key limitation in the Sajjan study, namely, that the use of ASMDM or cells alone when testing bacterial infection could not closely mimic the CF infection environment, as it lacks live host cells that would respond to the pathogen. First, they confirmed that *B. cenocepacia* K-56-2, a strain of Canadian origin [58] and A549 cells, was able to survive within the ASMDM environment, and this necessitated testing with increasing concentrations of ASMDM in HamF-12 cell maintenance media. Both bacteria and cells were able to survive within a maximum of 60% ASMDM (40% cell maintenance media) when exposed individually and together. Then, this 60% ASMDM concentration was used to maintain and observe *B. cenocepacia* infection, and it was found that inoculated *B. cenocepacia* associated more closely with the cell monolayer when 60% ASMDM (40% cell maintenance media) was used compared to infection in conventional cell media. Taken together, these results suggest that the viscosity and high mucin content of CF sputum enables *B. cenocepacia* to grow in and on the A549 cell layers, thus enhancing the feasibility of this model being used as a starting point for studying other opportunist CF species.

The Wijers et al. combination model could thus be utilised by researchers to provide insight into the progress of microcolony and biofilm infection under host-like conditions, to investigate other opportunistic infections in CF and the potential treatment kinetics in a replicable model.

## 3. Future Directions and Conclusions

The cystic fibrosis lung is a complex environment where the interplay between nutritional components in CF sputum and opportunistic bacterial infections needs to be balanced and understood prior to therapeutic intervention. Media formulations aiming to replicate CF sputum have been developed, but the variation in ingredients subsequently influences interpretation of experimental results. Considering the variation of formulations that exist, the first step in choosing a formulation is deciding on which one is the most comparable to CF sputum. The benefit of the SCFM media series lies in its specificity and similarities to extracted CF sputum, especially when compared with literature-based formulations. However, the first iteration of SCFM notably lacks the synthesis of purified mucin components despite being derived from the microanalysis of actual CF sputum. The first artificial sputum formulation developed by Ghani and Soothill was designed to observe mucoid phenotypes; however, today, there is a need to investigate numerous aspects of bacterial behaviour and how these contribute to bacterial fitness and treatment relevance. Subsequent changes to the original media were created to assist in assessing antimicrobial activity, microcolony formation, and genetic responses. By comparison, SCFM has been designed to not only assess mucoid phenotypes but also bacterial responses to specific nutritional cues. Any future formulations of SCFM media will need to be improved by the addition of BSA, protein-bound iron sources (e.g., ferritin), and bioactive lipids, all present in the CF lung environment and not included its current synthetic formulation.

In the investigative field, there is more need for media applications that focus on therapeutic interventions to reduce opportunistic bacterial loads. The study by Kostolowicz and Hall focused on creating models to study the diffusion of antibiotics through the mucus medium [50,52]. However, with the advent of antibiotic resistance observed in CF pathogens, the potential for combination therapies with novel agents is being investigated in vitro. In studies by Aiyer et al. (2021) which focus on the opportunistic pathogens *B.cenocepacia*, *Achromobacter xylosoxidans* and *Stenotrophomonas maltophilia* it was shown that the antioxidant *N*-acetylcysteine, in combination with the antibiotics colistin and ciprofloxacin, were able to reduce the bacterial load (CFU/mL) in vitro [59]. An extension of that study would be an assessment of the treatment kinetics of combination therapies (and separately, their components) through an artificial sputum to assess their delivery and efficacy in a replicable CF environment.

The CF lung model proposed by Wijers et al. (2016) is a promising pilot study where lung cells were added into artificial sputum media as opposed to conventional cell growth medium. While they were able to model bacterial infection and the relative tolerance of cells to artificial sputum, a limitation of the study was the use of the A549 cell line, which is an adenocarcinoma human alveolar basal epithelial cell line. Considering that opportunistic species would preferentially infect the bronchi rather than the alveoli, the use of a bronchial epithelial cell line, such as BEAS-2B or the ATCC approved cell line pair of NuLi-1 and CuFi-1, would be more beneficial to model the CF microenvironment. Future studies using a combined lung cell line–ASMDM media could be expanded by studying CF species of emerging significance such as *A. xylosoxidans* and *S. maltophilia*, especially as their prevalence is increasing due to longer life expectancy of CF patients.

The combined lung cell–ASMDM model could also be used in a study to assess the location of bacterial colonisation. If this is successfully achieved, it may be possible to infect the model with multiple bacterial colonisations and to use different tags for tracking each bacterial species. As the CF lung is rarely colonised by one species of bacterium, being able to follow multi-species colonisation would be of great importance in understanding the requirements for complex treatment options.

Artificial sputum media therefore represent not only a cost-effective and relatively simple method to model infections but can be adapted to suit the addition of host cell layers, thus providing an opportunity to conduct long-term adaptation experiments to monitor the effects of antibiotics or stressors on evolutionary divergence in bacterial populations, a common effect identified within the bacterial species inhabiting CF sputum.

## Figures and Tables

**Figure 1 microorganisms-10-01269-f001:**
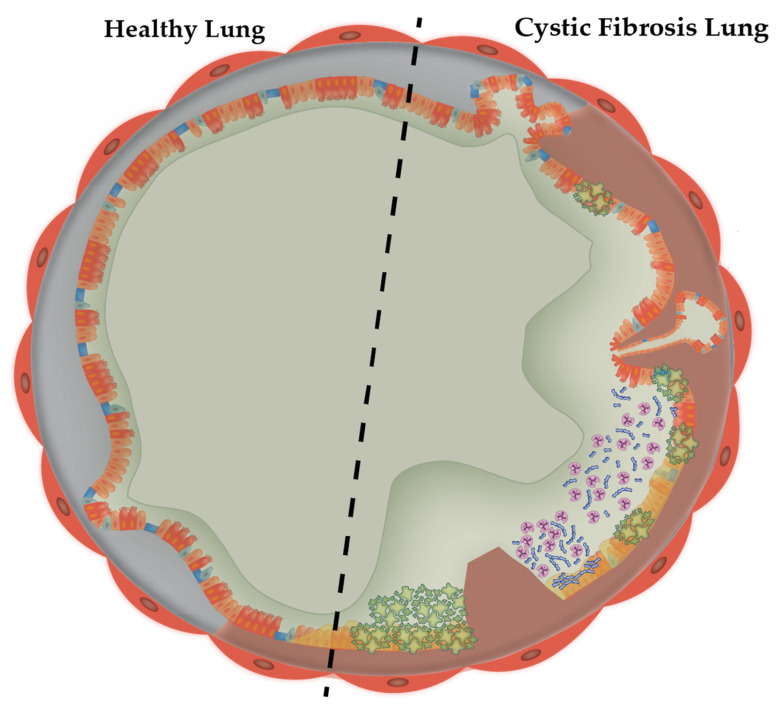
**CFTR dysfunction in cystic fibrosis bronchiole**. Healthy bronchiole airways (left of dotted line) are generally covered with a thin layer of mucus. Hydration of this layer is maintained by the cystic fibrosis transmembrane-conductance regulator and Cl^−^ channels. Mucus functions to trap pathogens and debris, eventually expelling it from airways via mucociliary escalation. However, in CF bronchioles (right), the CFTR is dysfunctional, resulting in disrupted ionic transport which causes hyper-concentrated and dehydrated mucus and subsequent osmotic compression of cilia (inset). This results in an inflammatory pathogenic cascade starting with mucus accumulation, mucus plug formation and opportunistic bacterial infections.

**Figure 2 microorganisms-10-01269-f002:**
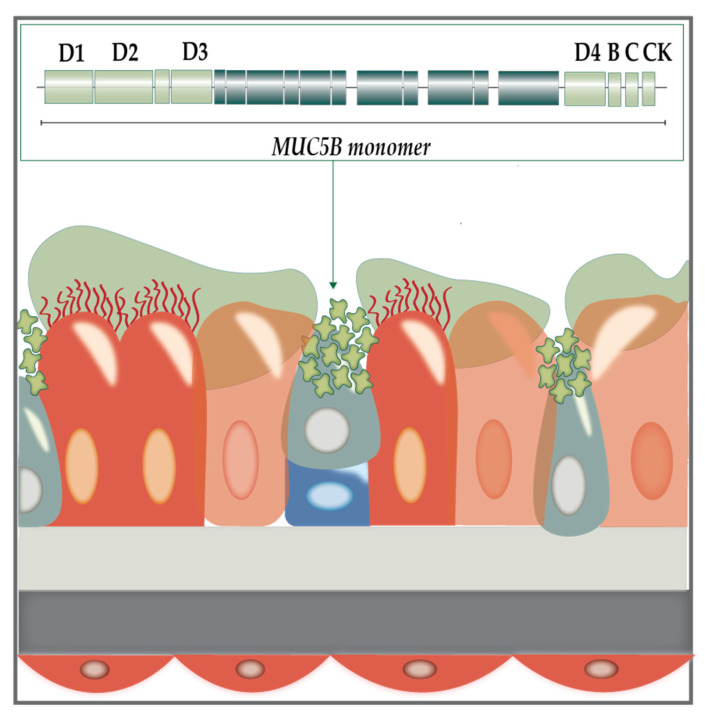
**Mucus hypersecretion in the CF lung**. CFTR dysfunction leads to mucus hypersecretion, which is key in CF lung pathophysiology. Secretory (orange), Clara (teal) and goblet cells (blue) are responsible for maintenance of the mucus gel layer, which is predominantly composed of MUC5B (arrowed above), and the similarly structured MUC5AC (not shown) monomers are connected via cross-linking. Each monomer is organised into N-terminal regions that contain N-N polymerisation von Willebrand factors D1-3 (light green), glycosylated mucin domains (dark blue) and C-terminal regions with the remaining von Willebrand factors: D4, B, C, and CK which coordinate C-C polymerisation [6,7]. The proportions of these homotypic polymers increase in response to inflammatory stimuli. Figure adapted from Fahy and Dickey (2010) [8].

**Figure 3 microorganisms-10-01269-f003:**
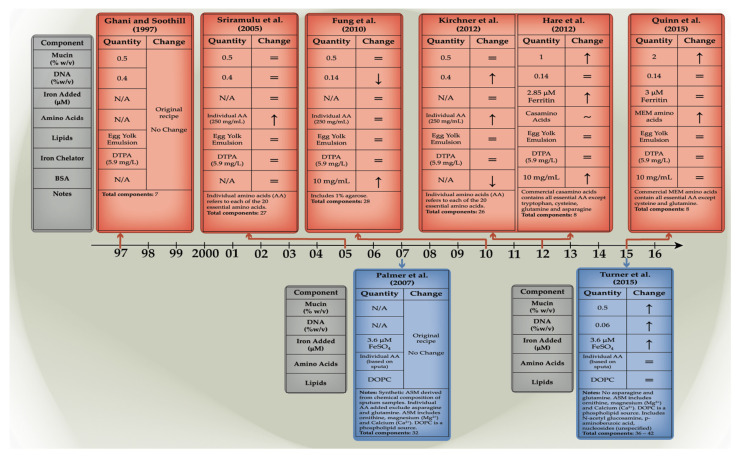
**Development timeline of artificial sputum formulations**. Formulations have been either derived from cited literature values (red) or are synthetically derived from CF sputum microanalysis (blue). Timeline adapted from Neve et al. (2021) [33]. References in figure from top left-right [17,22,32,34,35,36] and bottom left-right [15,37].

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
