# Peer review of "The Use of Artificial Sputum Media to Enhance Investigation and Subsequent Treatment of Cystic Fibrosis Bacterial Infections"

_microorganisms, 2022, doi:10.3390/microorganisms10071269_

Round 1
Reviewer 1 Report
This is an interesting review on artificial sputum media. It would be useful to those in this field and I can see it being well cited. There is a heavy focus on teh authors studies however, I would not say this is in appropriately self-cited. I have some suggestions for additional references below which I hope will strengthen the review. I ercommend that this should be published with the minor additions suggested.
Page 3, line 73-74: This makes it read as if biofilm formation takes a long time and only occurs well into chronic infection. Please reference a recent study by Kolpen et al (2022), Thorax on biofilms in acute and chronic lung infections.
Page 4, line 119: Sriramulu et al., 2005 highlighted the importance of amino acids in ASM. This reference should be added.
Line 375-394 – There is a very heavy focus on studies of the Australian epidemic strains. Many studies have now used sputum mimics either for expression, evolution or therapeutics. Perhaps some examples of different types of studies (including a range of clinical and reference strains) could be included.
Author Response
Response to Reviewer 1
Point 1: Page 3, line 73-74: This makes it read as if biofilm formation takes a long time and only occurs well into chronic infection. Please reference a recent study by Kolpen et al (2022), Thorax on biofilms in acute and chronic lung infections.
Response 1: We thank you for your comment. We did place an emphasis on chronic infection biofilms as the prime example of the eradication problem faced by CF patients, however we acknowledge that biofilms form at every stage of infection. We have rearranged the sentence to read as follows and have added in the Kolpen et al. (2022) reference.
Line 76-77: “…often followed by formation of microbial biofilms eventually culminating into recurrent and subsequently chronic infections [12-14]…”
Line 641: [14] Kolpen, M., et al., Bacterial biofilms predominate in both acute and chronic human lung infections. Thorax, 2022.
Point 2: Page 4, line 119: Sriramulu et al., 2005 highlighted the importance of amino acids in ASM. This reference should be added.
Response 2: We agree and have added in this reference to supplement the existing sentence in the manuscript
Line 125: “…matrix [24]. While…”
Line 660-661: [24] Sriramulu, D.D., et al., Microcolony formation: a novel biofilm model of Pseudomonas aeruginosa for the cystic fibrosis lung. J Med Microbiol, 2005. 54(Pt 7): p. 667-676.
Point 3: 375-394 – There is a very heavy focus on studies of the Australian epidemic strains. Many studies have now used sputum mimics either for expression, evolution or therapeutics. Perhaps some examples of different types of studies (including a range of clinical and reference strains) could be included.
Response 3: We understand the reviewers comments and we would like to highlight that the reason for this is that the Australian Epidemic strains were the focus of several artificial sputum studies and this is reflected in their mention. This includes the Fung et al (2010), Naughton et al. (2011). Palmer et al. (2007), and Klare et al (2016) studies
We would also like to point out that we have also highlighted other Pseudomonas aeruginosa strains used in other ASM studies, including the Liverpool Epidemic strains, and other non-Pseudomonas species all detailed below. These highlight the ability of ASM to be used for species other than Pseudomonas based studies and australian sourced strains.
To this end, we have added in the clinical and/or reference strain identifications for strains in the studies explored in the review.
Line 283-284: “The strains used included UCBPP-PA14, AES-1R and 2 clinical CF isolates…”
Line 327-328: “…(SCFM1), testing its efficacy using P.aeruginosa strain UCBPP-PA14 and Staphylococcus aureus strain Xen 36…”
Line 412: “…isolates, including AES-1 strains and Liverpool Epidemic strains (LES),…”
Line 433-434: “…using a PAO1 reference strain and a variety of LES strains with 6 extra non-LES strains,…”
Line 540: “…Firstly, they confirmed that B. cenocepacia K56-2, a strain of Canadian origin [60]…”
Line 727-728: [60] Garcia-Romero, I. and M.A. Valvano, Complete Genome Sequence of Burkholderia cenocepacia K56-2, an Opportunistic Pathogen. Microbiol Resour Announc, 2020. 9(43).
Reviewer 2 Report
The review of Aiyer and Manos about the use of ASM for the study of CF bacterial infections is interesting and gives an overview of the different formulations of ASMs and their results. This review also shows the ASM use for the development of antimicrobial treatment regimens and examination of bacterial interactions at the epithelial cell surface.
Minor comments:
Figure 2 - Is missing in legend the explanation of the upper part of the image (D1, D2, D3, D4, B, C, CK).
Figure 2 - If possible it will be interesting the adittion on the figure of MUC5AC.
Page 5, line 162 - Please remove dot after "nutrient".
Reference 43 - Please confirm if is correct and complete this citation.
Author Response
Point 1: Figure 2 - Is missing in legend the explanation of the upper part of the image (D1, D2, D3, D4, B, C, CK).
Response 1: We thank the reviewer for identifying this omission and we have added in description to the legend, as below
Lines 59-62: “…cross-linking. Each monomer is organised into N-terminal regions that contain N-N polymerisation von Willebrand factors D1-3 (light green), glycosylated mucin domains (dark blue) and C-terminal regions with the remaining von Willebrand factors factor D4, B, C, and CK which coordinate C-C polymerisation ([6, 7]). The…”
Line 629-630: [6] Thornton, D.J., K. Rousseau, and M.A. McGuckin, Structure and function of the polymeric mucins in airways mucus. Annu Rev Physiol, 2008. 70: p. 459-86.
Line 631-632: [7] Thornton, D.J. and J.K. Sheehan, From mucins to mucus: toward a more coherent understanding of this essential barrier. Proc Am Thorac Soc, 2004. 1(1): p. 54-61.
Point 2: Figure 2 - If possible it will be interesting the adittion on the figure of MUC5AC.
Response 2: We thank you for your comment, however the structures of MUC5B and MUC5A are incredibly similar and so for the sake of brevity and to avoid repetitiveness, the authors have chosen not to add the graphic of MUC5AC.
Point 3: Page 5, line 162 - Please remove dot after "nutrient".
Response 3: We have adjusted and removed it as per your request
Line 165: “…iron is an essential bacterial nutrient sensed by quorum signalling, for biofilm structural…”
Point 4: Reference 43 - Please confirm if is correct and complete this citation.
Response 4: We have rechecked this reference, and it is correct. Lopez reference is now Reference [45]. Lopez, M.J. and S.S. Mohiuddin, Biochemistry, Essential Amino Acids, in StatPearls. 2022: Treasure Island (FL).